

# Stabilizing effect of mélange buttressing on the Marine Ice Cliff Instability of the West Antarctic Ice Sheet

Tanja Schlemm[1,2], Johannes Feldmann[1], Ricarda Winkelmann[1,2], and Anders Levermann[1,2,3]

[1]Potsdam Institute for Climate Impact Research, Potsdam, Germany
[2]Institute of Physics and Astronomy, University of Potsdam, Potsdam, Germany
[3]Lamont-Doherty Earth Observatory, Columbia University, New York, USA

**Correspondence:** anders.levermann@pik-potsdam.de

**Abstract.** Due to global warming and particularly high regional ocean warming, both Thwaites and Pine Island glaciers in the Amundsen region of the Antarctic Ice Sheet could lose their buttressing ice shelves over time. We analyze the possible consequences using the Parallel Ice Sheet Model (PISM), applying a simple cliff-calving parameterization and an ice-mélange-buttressing model. We find that the instantaneous loss of ice-shelf buttressing, due to enforced ice-shelf melting, initiates
grounding line retreat and triggers the marine ice sheet instability (MISI). As a consequence, the grounding line progresses into the interior of the West Antarctic Ice Sheet and leads to a sea level contribution of 0.6 m within 100 a. By subjecting the exposed ice cliffs to cliff calving using our simplified parameterization, we also analyze the marine ice cliff instability (MICI). In our simulations it can double or even triple the sea level contribution depending on the only loosely constraint parameter which determines the maximum cliff-calving rate. The speed of MICI depends on this upper bound on the calving rate which
is given by the ice mélange buttressing the glacier. However, stabilization of MICI may occur for geometric reasons. Since the embayment geometry changes as MICI advances into the interior of the ice sheet, the upper bound on calving rates is reduced and the progress of MICI is slowed down. Although we cannot claim that our simulations bear relevant quantitative estimates of the effect of ice-mélange buttressing on MICI, the mechanism has the potential to stop the instability. Further research is needed to evaluate its role for the past and future evolution of the Antarctic Ice Sheet.

# 1  Introduction

Ice loss from the Greenland and Antarctic ice sheets is contributing increasingly to global sea level rise (Rignot et al., 2014; Shepherd et al., 2018b; WCRP Global Sea Level Budget Group, 2018). Ice sheets gain mass through accumulation of snowfall. Whether they contribute to sea level changes depends on how much this mass gain is offset or overcompensated by mass
losses due to surface and basal melting as well as iceberg calving. Both ice sheets in Greenland and Antarctica are currently losing ice (Enderlin et al., 2014; Shepherd et al., 2018b; Mouginot et al., 2019; Larour et al., 2019; Bell and Seroussi, 2020). Estimating the additional future mass loss of these ice sheets is critical for future sea level projections (Church et al., 2013; Ritz



et al., 2015; DeConto and Pollard, 2016; Mengel et al., 2016; Kopp et al., 2017; Slangen et al., 2017; Golledge et al., 2019; Levermann et al., 2020; Edwards et al., 2021). Uncertainties in modeling the physics of the Antarctic Ice Sheet (AIS) lead to large uncertainties in sea level projections (Noble et al., 2020; Pattyn and Morlighem, 2020).

One such uncertainty is the potential collapse and the calving of large ice cliffs after the ice shelves buttressing them have disintegrated. The concept of cliff calving was motivated by an analysis of depth-averaged stresses near an ice cliff, which showed that ice cliffs exceeding an ice thickness stability limit are inherently unstable (Bassis and Walker, 2011). Cliff calving could lead to uncontrolled ice retreat: Grounding line retreat caused by cliff calving may expose even higher ice cliffs further inland, which in turn are more susceptible to collapse, resulting in self-reinforcing ice retreat. This is referred to as Marine Ice

Cliff Instability (MICI).
A study by DeConto and Pollard (2016) found that the AIS could contribute up to 1 m of sea level rise within a century, if cliff calving is taken into account. This is substantially more than all other projections that do not include MICI. However, this study has been criticised as over-estimating sea level contribution (Edwards et al., 2019) due to a lack of observationally constrained models of the cliff calving process. DeConto and Pollard (2016) parameterized cliff calving with a step-like function that is

zero for ice cliffs below the stability limit and ramps up rapidly to an upper limit for all ice cliffs exceeding the stability limit. We revisit the question of MICI in the AIS using a more complex parameterization of cliff calving, which is based on the shear failure of an ice cliff and gives the cliff calving rate as an exponential function of ice thickness and water depth (Schlemm and Levermann, 2019). A recent, more detailed modeling study of ice cliff failure, incorporating different structural failure modes as well as surface lowering due to viscous deformation, supports the findings that calving rates increase exponentially with ice

thickness (Crawford et al., 2021). In our model, we further assume that calved icebergs form an ice mélange that buttresses the ice cliffs, providing an upper bound on calving rates (Schlemm and Levermann, 2021).

We consider the Amundsen region of the West Antarctic Ice Sheet (WAIS) as the likely initiator of MICI. Iceberg plow marks on the seafloor indicate that large full thickness icebergs calved from Pine Island Glacier and that MICI was active in

this area during the last deglaciation (Wise et al., 2017). Additionally, the WAIS is grounded largely on bedrock below sea level and is therefore vulnerable to both the Marine Ice Sheet Instability (MISI) and MICI. MISI is caused by grounding line retreat on a retrograde bed: Retreat into deeper bed regions increases the flux across the grounding line and therefore accelerates grounding line retreat, resulting in self-reinforcing ice loss (Mercer, 1978; Schoof, 2007; Favier et al., 2014). Observations show that MISI is possibly already underway in the Amundsen region (Joughin et al., 2014; Mouginot et al., 2014; Rignot

et al., 2014). Once MISI is initiated, the entire WAIS could collapse on a millennial time scale, resulting in sea level rise of 3 m (Feldmann and Levermann, 2015). With the addition of cliff calving (MICI), the WAIS collapse would occur much more rapidly.

The breakup of ice shelves is a necessary precondition for the calving of exposed ice cliffs and thus for the onset of MICI.

Hydrofracturing, in which the deepening of ice crevasses due to extensive surface meltwater leads to the catastrophic failure



of an entire ice shelf, has been proposed by DeConto and Pollard (2016) as the main mechanism for ice shelf breakup and the consequent exposure of ice cliffs.

In 2002, the Larsen B ice shelf on the Antarctic Peninsula collapsed within a week after having thinned in previous years due to high summer melt rates (Rack and Rott, 2004; Glasser and Scambos, 2008). As a result of the ice shelf collapse, glaciers
flowing into the shelf have permanently accelerated (Scambos et al., 2004; Berthier et al., 2012). These are small glaciers with little impact on the overall Antarctic mass balance. Based on the observation of numerous surface meltwater ponds prior to ice shelf collapse, it has been suggested that hydrofracturing due to intense surface melting was the primary cause of this sudden collapse (MacAyeal et al., 2003). However, anomalously large surface melt rates are required for an ice shelf to break up as rapidly as the Larsen B ice shelf did (Robel and Banwell, 2019). Thus, hydrofracturing would probably not be the main mech-
anism leading to ice shelf failure in the Amundsen region: Even under the RCP 8.5 scenario, surface meltwater production on the Pine Island ice shelf is projected to remain far below a threshold of 300 mm/a at the end of the century (Trusel et al., 2015). This threshold is equivalent to current surface meltwater production on the remaining Larsen C ice shelf and less than half of the pre-collapse surface meltwater production on the Larsen B ice shelf (Trusel et al., 2015). Therefore, it is unlikely that the ice shelves in the Amundsen region will fail due to hydrofracturing.

Nevertheless, it is likely that the ice shelves in the Amundsen region will break apart under persisting global warming conditions. The Amundsen Sea is warming (Shepherd et al., 2004, 2018a), leading to increased basal melting of ice shelves. This is already causing thinning and grounding line retreat in all the glaciers in the Amundsen region (MacGregor et al., 2012; Mouginot et al., 2014; Milillo et al., 2019).
The destabilizing effect of basal melt on ice shelves can be further amplified by crevasses: Satellite observations show a trend of widespread rifting at the shear margins of all glaciers in the Amundsen region (MacGregor et al., 2012) as well as an increase in rifts originating from basal crevasses in the center of the Pine Island ice shelf (Jeong et al., 2016). Ocean warming may be the cause of the observed expansion of basal crevasses (Jeong et al., 2016). Rifting and crevassing accelerates grounding line retreat: Damage feedback modeling showed that a basal melt rate of 20 m/a combined with a 20 m deep crevasse in the
shear zone at the grounding line causes a faster grounding line retreat than a basal melt rate of 100 m/a on an undamaged shelf (Lhermitte et al., 2020).

In addition, calving front retreat of small ice shelves may be self-reinforcing: a linear elastic fracture mechanics model of calving at Thwaites Glacier showed a positive feedback, i.e., if calving results in a shorter ice shelf, this shorter ice shelf is more likely to calve (Yu et al., 2017). It is also possible that weakened buttressing due to ice shelf thinning at Pine Island
and Thwaites glaciers could amplify the development of damage in their shear zones. Lhermitte et al. (2020) suggest that this damage feedback may predispose the ice shelves at Pine Island and Thwaites glaciers for disintegration. This would remove buttressing from glaciers terminating in the Amundsen Sea and expose large ice cliffs, triggering MISI and MICI.

We perform a series of simulations using the Parallel Ice Sheet Model (PISM) in a regional setup of the WAIS, where we
initiate MISI and MICI by removing the ice shelves in the Amundsen region. The ice sheet model and calving parameterizations





are described in more detail in sec. 2. We present the resulting sea level contributions in sec. 3. In sec. 4, we discuss how the strength of mélange buttressing changes with grounding line retreat and show that as a result MICI slows down as it progresses.

## 2 Methods

### 2.1 PISM

We carry out regional simulations of the WAIS with PISM (Bueler and Brown, 2009; Winkelmann et al., 2011) at a horizontal resolution of $4\,\mathrm{km}$. The model setup is similar to the one used and described in Feldmann et al. (2019).

PISM is a thermomechanically coupled model based on the Glen–Paterson–Budd–Lliboutry–Duval flow law (Aschwanden et al., 2012). It uses a superposition of the shallow ice approximation (Hutter, 1983) and the shallow shelf approximation (Morland, 1987; MacAyeal, 1989), allowing for a smooth transition between different ice sheet flow regimes. Basal friction

is calculated using a nonlinear Weertman-type sliding law with a sliding exponent of 3/4 combined with a Mohr-Coulomb model for plastic till (Bueler and van Pelt, 2015) that accounts for the effect of evolving ice thickness and the associated change in overburden pressure on the basal till. The till friction angle is parametrized with bed elevation. This friction scheme ensures a continuous transition from quasi–nonslip regimes in elevated regions to the marine areas where basal resistance is low. The grounding line position is free to evolve using hydrostatic equilibrium. Grounding line movement has been evaluated

in the model intercomparison projects MISMIP3d (Pattyn et al., 2013; Feldmann et al., 2014) and MISMIP+ (Cornford et al., 2020). Basal friction at the grounding line is interpolated according to a sub-grid, linear interpolation of the grounding line position (Feldmann et al., 2014). Basal melt rates under ice shelves are calculated using the Potsdam Ice-shelf Cavity mOdel (PICO) (Reese et al., 2018a), where ocean conditions are determined by mean values over the observational period 1975-2012 (Schmidtko et al., 2014). The surface mass balance and ice surface temperature are averaged from RACMO2.3p2 1986-2005

(van Wessem et al., 2018).

The model domain includes the West Antarctic Ice Sheet, the Antarctic Peninsula and parts of the East Antarctic Ice Sheet, in particular the drainage basins towards Ross and Ronne-Filchner ice shelves (Zwally et al., 2012). The bed topography and initial ice configuration were taken from Bedmap2 (Fretwell et al., 2013).

### 2.2 Calving laws

#### 2.2.1 Breakup of ice shelves

In our simulations, we assume that in the near future the ice shelves in the Amundsen region will break apart and will not be able to regenerate. This is a very strong assumption and is implemented in PISM with a so-called 'floatkill' mechanism, which removes all floating ice in the Amundsen region at each time step.

For the remaining ice shelves, mainly the Ross and Ronne-Filchner ice shelves, but also small ice shelves along the Antarctic Peninsula, the so-called eigencalving parameterization is applied (Levermann et al., 2012).



### 2.2.2 Mélange-buttressed cliff calving

For the ice cliffs, i.e. grounded ice sheet at the coast, we use a cliff-calving relation based on shear failure of an ice cliff (Schlemm and Levermann, 2019). If the difference between ice thickness and water depths lies below a water depth dependent threshold ($\approx 100\,\mathrm{m}$), the cliff is assumed to be stable. For larger ice cliffs, the calving rate grows exponentially with ice thick-
ness and water depth.

This assumed exponential relation and the fact that in many regions in West Antarctica the bed topography is down-sloping inland, can lead to very large calving rates ($> 30\,\mathrm{km/a}$, see fig. 1a). In addition to the recently discussed stabilizing effect of dynamic thinning (Bassis et al., 2021; Golledge and Lowry, 2021), a mélange of icebergs and sea ice, may have a stabilizing effect on MICI. Here we apply a very simple mélange-buttressing parameterization (Schlemm and Levermann, 2021). Larger calving rates lead to the production of more icebergs, which together with sea ice form a stiff ice mélange. This mélange buttresses the ice cliff, thereby stabilizing it. As a result of this negative feedback between calving rate and mélange buttressing, there is an upper limit to the calving rate, $C_{max}$ (see fig. 1b). This threshold, derived in Schlemm and Levermann (2021), is a function of embayment geometry and mélange properties,

$$C_{max} = \frac{W_{ex}}{W_{cf}} \left( b_0 + b_1 \mu_0 \frac{L_{em}}{W_{em}} \right)^{-1} \gamma \, u_{ex} \, ., \tag{1}$$

where the mélange length is denoted by $L_{em}$, the mélange width at the calving front by $W_{cf}$, the mélange exit width by $W_{ex}$ and the average mélange width by $W_{em}$ (see fig. 2). $\gamma$ is the fraction of the ice thickness $H$ beyond which calving is completely suppressed, and $u_{ex}$ is the exit velocity, with which mélange drifts out of the embayment. Finally, the internal friction of the mélange, $\mu_0$, has values between 0.1 and 1, and the linearization parameters are given by $b_0 = 1.17$ and $b_1 = 1.11$. We chose the mélange parameters as $\mu_0 = 0.3$, $\gamma = 0.2$ and $u_{ex} = 100\,\mathrm{km/a}$ (for a discussion on the value of the exit velocity, see sec. 4.1.1). The value of $C_{max}$ then depends solely on the embayment geometry (fig. 2).

In order to estimate $C_{max}$ for a given grounding line configuration, we assume that the entire embayment is filled with mélange. We can then estimate the width of the mélange exit, $W_{ex}$, and the length of the calving front, $W_{cf}$. The average mélange width, $W_{em}$, is calculated as the average of $W_{ex}$ and $W_{cf}$. The mélange length, $L_{em}$, is calculated as the average distance between the embayment exit and the calving front (the resulting trapezoids are shown in fig. 11b). Tab. 1 shows estimates of $C_{max}$ for Thwaites and Pine Island glaciers as well as for two extreme cases of mélange geometry: a narrow and long mélange strongly buttresses the calving front, resulting in a small $C_{max}$, while a wide and short mélange provides little buttressing at the calving front, resulting in a large $C_{max}$.

Similar to the 'floatkill' parameterization, mélange-buttressed cliff calving is not applied to the entire model domain, but only to the coast of the Amundsen region and the interior of the WAIS. This implementation prevents MISI and MICI from starting in other regions of the AIS. The shaded region in fig. 5 shows the region where the 'floatkill' parameterization and

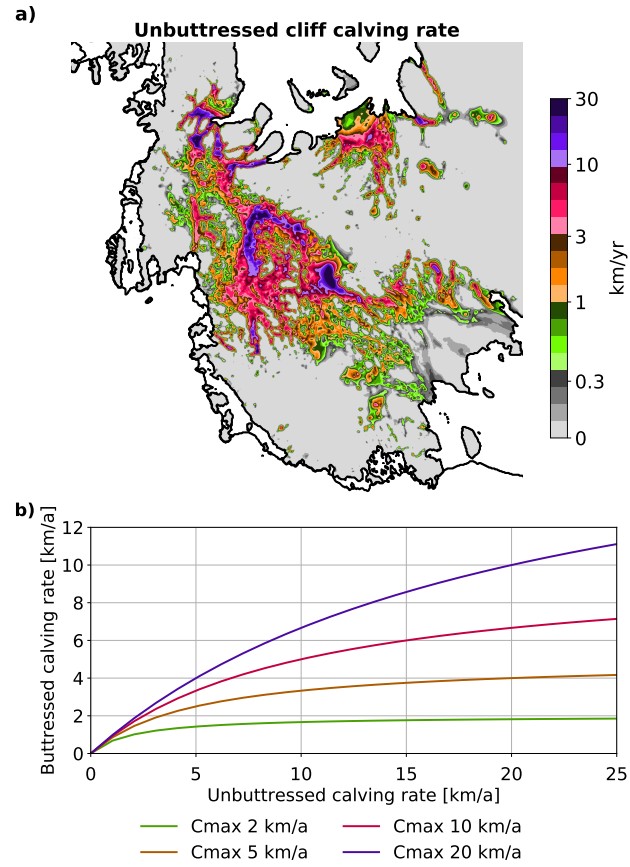

**Figure 1.** a) Potential unbuttressed cliff calving rates in the WAIS. For this estimate we assume the ice cliff to be at floatation thickness, making the calving rate a function of bed topography. In the case of very fast grounding line retreat, the ice cliff may not have thinned to floatation and calving rates may be larger. b) The mélange-buttressed calving rates as a function of the unbuttressed calving rates for the values of $C_{max}$ considered in this study.

mélange-buttressed cliff calving are not applied.

## 2.3 Initialisation and experiments

The ice sheet was spun up into thermal equilibrium with fixed bed and ice geometry. A further 10-year run with evolving ice
5 geometry was performed to remove short-lived floating regions in the WAIS (such as in the middle of Smith glacier, west of Thwaites glacier).

Five types of experiments were carried out:

REF: a reference simulation with current day atmosphere and ocean conditions held constant,



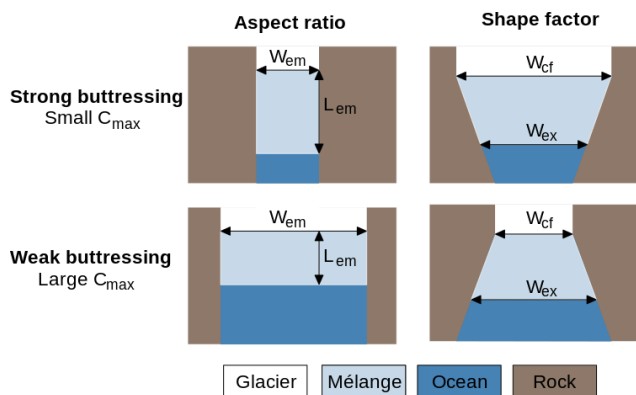

**Figure 2.** Illustration of how embayment geometry determines buttressing strength in eq. 1: Aspect ratio $L_{em}/W_{em}$ and shape factor $W_{ex}/W_{cf}$ determine the strength of mélange buttressing.

**Table 1.** Upper bounds on calving rates given by eq. 1 with $\mu_0 = 0.3$, $\gamma = 0.2$ and $u_{ex} = 100\,\text{km/a}$. We first consider two extremes of a narrow and long as well as a wide and short buttressing mélange, while assuming a rectangular mélange geometry with constant mélange width, $W_{ex} = W_{cf} = W_{em}$. For Thwaites and Pine Island glacier, we assume mélange geometry similar to the current ice shelf. The smaller the upper bound $C_{max}$, the stronger the buttressing effect caused by the ice mélange.

|  | $W_{em}$ [km] | $L_{em}$ [km] | $W_{ex}/W_{cf}$ | $C_{max}$ [km/a] |
|---|---|---|---|---|
| narrow and long | 5 | 100 | 1 | 2.6 |
| wide and short | 200 | 5 | 1 | 17.0 |
| Thwaites Glacier | 93 | 14 | 1.19 | 19.6 |
| Pine Island Glacier | 48 | 58 | 1.14 | 15.5 |

BMT: the 'basal melt experiment' is a melt experiment with current day atmospheric conditions and the melt rate in the Amundsen basin set to 200 m/a,

FLK: the 'floatkill'-parameterization experiment with current day atmospheric and ocean conditions, in which all floating ice in the Amundsen basin and the interior of the WAIS was removed,

5  CC#: the four cliff calving experiments, which were performed in the same way as the the 'floatkill'-parameterization experiment, with the addition of exposing grounded glacier margins to cliff calving with different upper limits, $C_{max} = [2, 5, 10, 20]$ km/a (CC2, CC5, CC10, CC20),

CCA: a fifth cliff calving experiment with an adaptive upper bound $C_{max}$, which was updated every 5 model years for the new embayment geometry.





Each experiment was run for 100 a. The the 'floatkill'-parameterization experiment and the three cliff calving experiments with small $C_{max}$ as well as the adaptive cliff calving experiment were extended until they reached a retreat comparable to the fastest cliff calving experiment with $C_{max} = 20$ km/a.

## 3 Results

To compare MISI and MICI with different upper bounds, we use the contribution to sea level over time as a measure for the speed of the instabilities. In the cliff calving experiments, MICI occurs in addition to MISI. Even though the two instabilities are very likely to interact we choose to estimate the contribution of MICI by measuring the sea level contribution difference between the 'floatkill'-parameterization (FLK) and the cliff-calving (CC#) experiments. Thus, for each cliff-calving experiment, we refer to the MISI contribution as to the discharge from the FLK experiment.

### 3.1 MISI discharge caused by 'floatkill' is similar to that caused by basal melt

In our setup, the two MISI experiments (FLK and BMT) contribute about 0.6 m of sea level rise within 100 a (see fig. 3 and tab. 2). This corresponds to the upper limit of the sea level contribution from the Amundsen sector found in LARMIP-2 (Levermann et al., 2020), where a basal melt anomaly of up to 16 m/a was applied to currently observed melt rates. It is at the upper end of the 16 models that participated in LARMIP-2, but is not the highest.

The sea level contributions resulting from the FLK and BMT experiments are very similar. This agrees with results from the ABUMIP intercomparison study (Sun et al., 2020), which showed that Antarctic-wide ice loss due to large basal melt rates is comparable with ice loss due to the 'floatkill' parameterization.

The assumed basal melt rate in BMT is higher than the current and projected average melt rates of the Amundsen region ice shelves (Naughten et al., 2018). However, close to the grounding line of Thwaites glacier, basal melt rates of up to 200 m/a were

found (Milillo et al., 2019). In the melt experiment, this rate was applied to the whole of the ice shelves in the Amundsen region.

### 3.2 MICI discharge is controlled by upper bound on calving rates

For the two lowest upper bounds on cliff calving ($C_{max} = 2$ km/a and $C_{max} = 5$ km/a), MICI contributes a factor of up to 1.5 additionally to sea level rise from the MISI experiments (fig. 3 and tab. 2). For larger upper bounds, MICI can more than

double (CC10 with $C_{max} = 10$ km/a) or even triple (CC20 with $C_{max} = 20$ km/a) the sea level contribution compared to the MISI experiments (FLK, BMT). The sea level contribution of the adaptive experiment after 100 a lies between that of the CC5 and CC10 experiments (fig. 3 and tab. 2).

Ice retreat rates increase with time, with sea level rates for the FLK and CC2 experiments reaching about 1 mm/a after 100 a, while the CC20 experiment reaches its maximum sea level rate of 2.5 mm/a already after 50 a. The sea level rate of the CC20

experiment decreases after 60 a of runtime because the grounding line retreat along the Pine Island Glacier towards the Ronne Ice Shelf has reached the boundary of the inner WAIS region where cliff calving and the 'floatkill' parameterization are applied

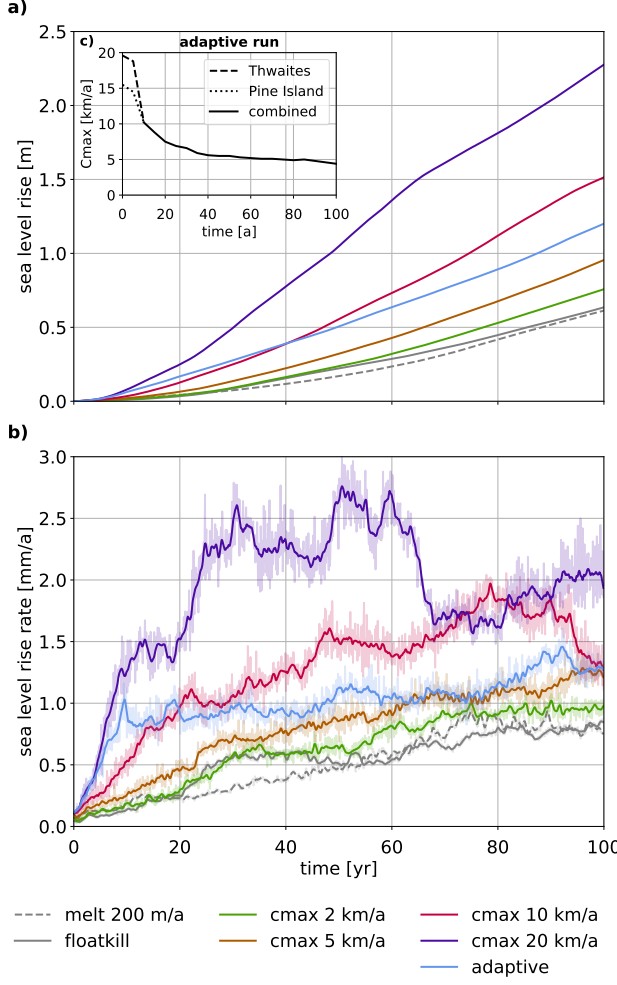

**Figure 3.** Cumulative sea level contribution (a) and rate of sea level rise (b) relative to the reference run for all six different experiments carried out. The curves for different $C_{max}$ are computed as the difference between the CC# simulations and the FLK experiment, i.e. they are estimates of the additional contribution from MICI over MISI. Shown are monthly data and a yearly average calculated with the LOESS (locally estimated scatterplot smoothing) algorithm (Cleveland, 1979). The inset (c) shows the upper bound on calving, $C_{max}$, of the adaptive experiment.

(see fig. 5).

In the adaptive experiment, the sea level rate increases similarly to the CC20 experiment for the first 10 a and then levels off, crosses the sea level rate of the CC10 experiment after about 20 a and finally approaches the sea level rate of the CC5 experiment. This corresponds to the reduction of the adaptive upper bound on calving rates, which initially starts at $C_{max} = 20$ km/a

5 for the Thwaites embayment, decreases to $C_{max} = 10$ km/a when the Thwaites and Pine Island glacier embayments merge,



**Table 2.** Sea level contribution after 50 a and 100 a. Cumulative calving discharge from the Amundsen region after 100 a and average calving amplification calculated as fraction between overall calving discharge and calving discharge due to the 'floatkill' parameterization for all MISI and MICI experiments performed. MISI is computed as the difference of BMT and FLK to the REF simulation. MICI is computed as the difference between CC#/CCA and FLK as described in the text.

| | | sea level contribution [m] | | cumulative | average calving |
|---|---|---|---|---|---|
| | | 50 a | 100 a | discharge [$10^6$ Gt] | amplification |
| MISI | BMT | 0.17 | 0.61 | - | - |
| | FLK | 0.22 | 0.64 | 4.00 | 1 |
| MICI | CC2 | 0.24 | 0.76 | 4.72 | 1.34 |
| | CC5 | 0.32 | 0.95 | 6.00 | 1.86 |
| | CC10 | 0.56 | 1.51 | 9.68 | 2.39 |
| | CC20 | 1.05 | 2.28 | 14.53 | 3.15 |
| | CCA | 0.51 | 1.20 | 7.64 | 2.02 |

and further approaches $C_{max} \approx 5$ km/a. (see fig. 3c).

Calving is the main cause of sea level rise: for experiments CC2 and CC5, the cumulative calving discharge is only slightly larger than for the FLK experiments; for experiments CC10 and CC20 the calving discharge doubles and triples, respectively. The slowdown of the CC20 experiment after 60 a is also visible in the reduced calving discharge. Similar to the sea level rate, the calving discharge of the adaptive experiment initially parallels that of the CC20 experiment, then levels off and roughly follows the calving discharge of the CC5 experiment (see fig. 4 and tab. 2).

We calculate the calving amplification as the ratio between the total calving discharge in the individual CC# experiments and the discharge in the FLK experiment (tab. 2). It reveals a doubling/tripling in the calving discharge for the highest values of $C_{max}$, similar to the increase in the sea level contributions mentioned above.

The cliff calving experiments with a small upper bound show only a modestly faster ice retreat compared to the 'floatkill' experiment. This may partly be due to the resolution: PISM uses a subgrid scheme for the ice margin, involving partially filled cells that are not affected by either the ice dynamics or the 'floatkill' mechanism. Cliff calving with a small value of $C_{max}$ can prevent partially filled cells from filling up and thus reduce the ice loss due to the 'floatkill' parameterization. This results in a slightly lower overall calving discharge. Cliff calving with a large value of $C_{max}$ is much more likely to completely remove partially filled cells, so the 'floatkill' parameterization mechanism is not hindered in this case.

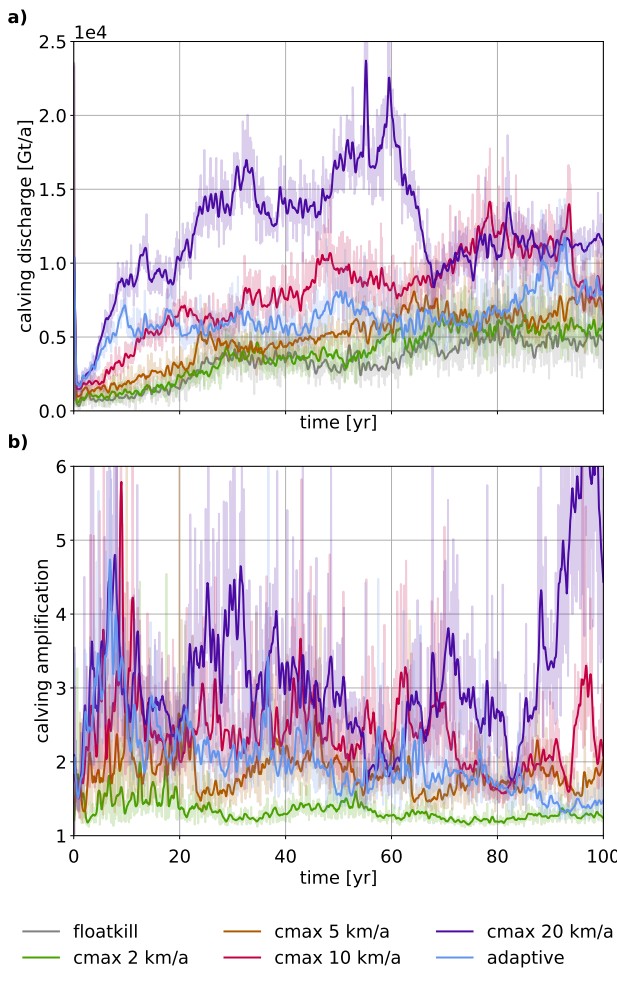

**Figure 4.** a) Overall calving discharge from the Amundsen region. b) The calving amplification calculated as the fraction between overall calving discharge and calving discharge due to the 'floatkill' parameterization only. Note that no calving amplification has been calculated for the 'floatkill'-only experiment because no cliff calving takes place. Shown are monthly data and a yearly average calculated with a LOESS algorithm.

### 3.3 Mélange buttressing increases as MICI progresses, slowing MICI speed

In the adaptive cliff calving experiment (see blue lines in fig. 3, 4 and 5), mélange buttressing strength depends on the embayment geometry (see eq. 1 and fig. 2). Because the embayment becomes wider and the calving front becomes longer, the upper bound on calving rate decreases with grounding line retreat into the Amundsen basin.

5  Initially, Thwaites and Pine Island glacier have separate embayments with different values for $C_{max}$. After 10 a, the embayments merge, leading to one value of $C_{max}$ for the whole Amundsen basin. The development of the upper bound with simulation time is shown in fig. 3c. In fig 6, the upper bound is shown as a function of the sea level contribution of the corre-





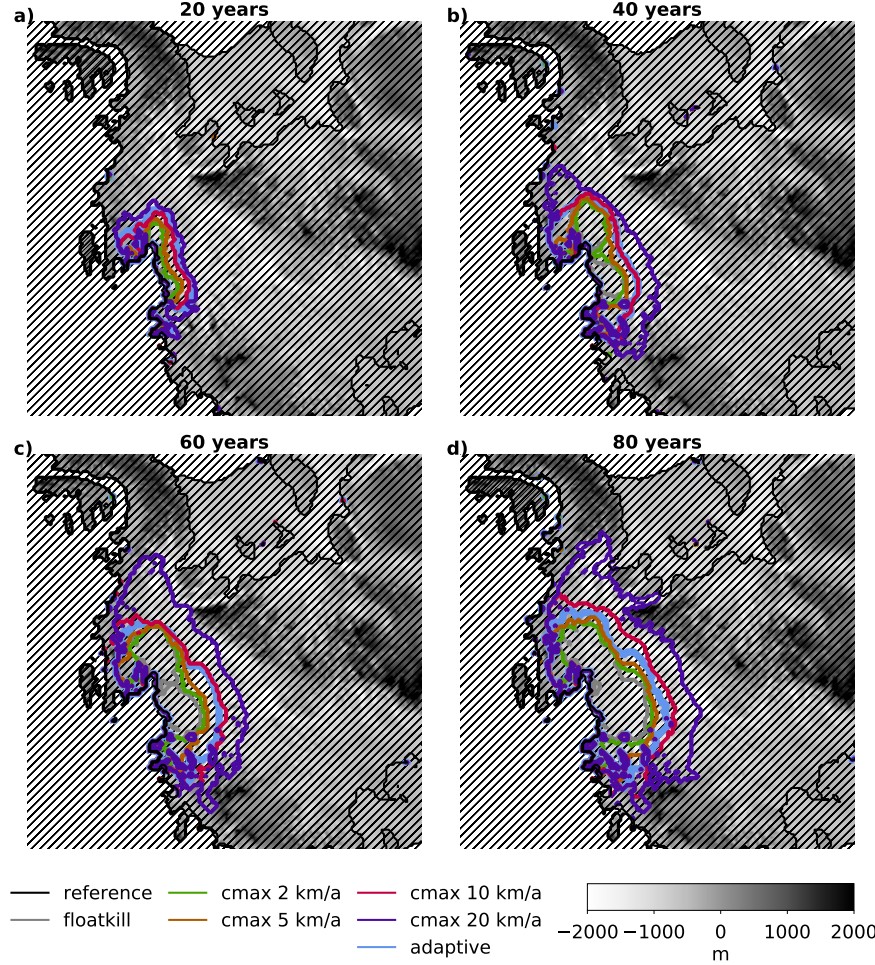

**Figure 5.** Maps of grounding line retreat in the WAIS, underlaid with the bed topography. In the shaded region, neither the 'floatkill' parameterization nor cliff calving is applied.

sponding embayment geometry. $C_{max}$ decreases from initially $\approx 20\,\text{km/a}$ for Thwaites glacier and $\approx 15\,\text{km/a}$ for Pine Island glacier to $\approx 5\,\text{km/a}$ as the grounding line retreats deeper into the Amundsen basin. The relation can be fitted with:

$$C_{max} \approx 3.92\,\text{km/a} \cdot \exp\left(\frac{0.15\,\text{m}}{\text{SLR} + 0.10\,\text{m}}\right). \tag{2}$$

5   As MICI progresses and the grounding line retreats, the area covered by ice mélange grows, which increases the strength of mélange buttressing. This in turn lowers the upper limit on calving rates and slows further progression of MICI. Thus, as a consequence of mélange buttressing, MICI cannot be arbitrarily fast and even decelerates as it progresses.



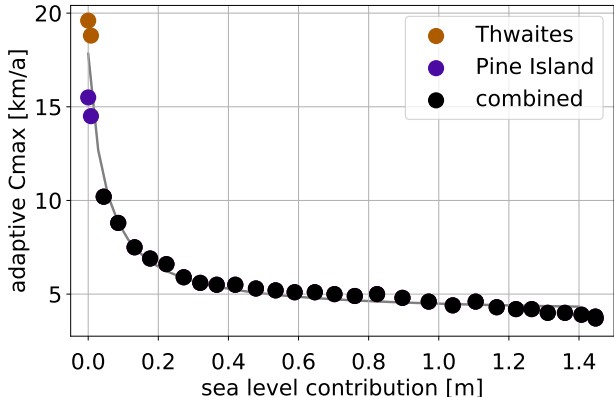

**Figure 6.** The upper bound on calving rates, $C_{max}$ of the adaptive cliff calving experiment as a function of the sea level contribution of the corresponding embayment geometry. Initially, Thwaites and Pine Island glacier have separate embayments, which merge after $10\,a$. The upper bound decreases with sea level contribution and with the corresponding simulation time (see fig. 3c) . The fit (eq. 2) is shown in grey.

### 3.4 Bed topography controls rate of grounding line retreat

The grounding line retreat initially follows the main flow directions of Pine Island and Thwaites glaciers, but after some time (depending on $C_{max}$) it involves the entire interior of the WAIS (see fig. 5). The retreat reaches the Ronne Basin earlier than the Ross Basin. The CC20 experiment reaches the Ronne Ice Shelf after $70\,a$ of runtime, where the retreat ends as no further

the 'floatkill' parameterization and cliff calving is allowed there. The retreat towards the Ross Ice Shelf continues. The experiments with smaller $C_{max}$ as well as the FLK experiment take longer to reach the Ronne Ice Shelf, with the FLK experiment being the slowest, arriving there after $150\,a$.

   We examine the retreat along two flowlines, leading from Thwaites glacier across to Ross ice shelf and from Pine Island

glacier across to Ronne ice shelf, respectively (see fig. 7): Both glaciers have retrograde beds, with Thwaites glacier having a steeper slope than Pine Island glacier. After the flowlines cross the initial ice divide, the bed topography changes: The retreating grounding line of Thwaites Glacier meets the Bindschadler Ice Stream, which has a rather shallow and slightly prograde bed topography (in the direction of grounding line retreat). In contrast, the retreating grounding line of Pine Island glacier reaches the Evans Ice Stream, which has a deep bed depression. Fig. 8 shows the retreat of the grounding line and ice divide over time.

For Thwaites glacier, all experiments show some inertia to the retreat initially, which is followed by rapid retreat along the first $150\,km$ of the flowline. Retreat then levels off, with experiments with larger $C_{max}$ showing faster retreat. Pine Island glacier shows steady initial retreat over the first $300\,km$, after which the retreat stalls for $25\,a$ to $50\,a$, depending on the experiment. This is followed by a rapid retreat that is stopped only when the grounding line reaches the Ronne Ice Shelf, where no further retreat is possible. As the grounding line retreats, so does the ice divide, but with a considerable delay.





An explanation for this retreat pattern can be found by a more detailed analysis that compares the grounding line retreat rates with the slope of the bed topography (see fig. 9). Grounding line retreat along the Thwaites flow line is rapid at first, with retreat rates up to $18\,\text{km/a}$ (depending on $C_{max}$) along a steep retrograde bed, and slows down once the grounding line reaches a more even bed topography segment beginning at $150\,\text{km}$. In this segment, retreat rates fluctuate below $10\,\text{km/a}$. Ridges in

the bed topography at $220\,\text{km}$ and $430\,\text{km}$ cause stagnation of grounding line retreat on the upslope, followed by acceleration on the downslope. A steady retrograde slope between $500\,\text{km}$ and $630\,\text{km}$ causes grounding line retreat rates to increase up to $10\,\text{km/a}$. The steep prograde slope between $630\,\text{km}$ and $700\,\text{km}$ causes the retreat to slow down significantly.

The retreat along the Pine Island flow line has a steady rate between $5\,\text{km/a}$ and $15\,\text{km/a}$ for the first $300\,\text{km}$ until the grounding line approaches a bathymetric ridge, where the retreat slows temporarily. A short $20\,\text{km}$ long depression following this ridge

causes an acceleration of up to $10\,\text{km/a}$, followed by a slowdown as the bed rises again. Grounding line retreat accelerates sharply up to values between $15\,\text{km/a}$ and $33\,\text{km/a}$ once it reaches a steep bed depression beneath Evans ice stream, which begins at $450\,\text{km}$ .

We expect bed topography to control grounding line retreat for two reasons: analytical calculations in a depth-averaged

flowline model show that the flux across the grounding line scales superlinearly with ice thickness (Schoof, 2007). The cliff calving rate also scales superlinearly with ice thickness (Schlemm and Levermann, 2019). Assuming that the glacier terminus is at floatation, this means that there should also be a relationship between the grounding line retreat rate and the bed depth. However, a correlation analysis using the Spearman correlation coefficient of determination between grounding line retreat rate and ice thickness shows only a minimal correlation for Pine Island Glacier and no correlation at all for Thwaites Glacier (see

tab. 3). The adaptive experiment has the smalles correlation coefficients because the upper bound on calving rates decreases as the grounding line retreats. For both glaciers, the correlation coefficients are even smaller, if we consider bed topography rather than ice thickness.

There are two main reasons for this: First, we analyze flow along a 1d flowline embedded in a more complex 2d flow. The retreat of the grounding line in neighboring flowlines, where the bed topography can be different, may drag on the grounding

line and either accelerate or decelerate it, in comparison to the result of the 1d analysis. In addition, the analyzed flowlines may not lie exactly along the flow direction, especially in the vicinity of bed topography disturbances that are only a few grid cells in size. Second, ice flow has inertia, which means that the grounding line takes some time to accelerate when it reaches a steep retrograde bed. Inertia can also drive it over bumps in the bed that would be expected to slow it down, especially in the case of large $C_{max}$.


In summary, we find no clear statistical correlation between the bed topography and the grounding line retreat rate. Nevertheless, we observe an acceleration of the grounding line when the bed is retrograde and a deceleration when it is prograde. In addition, bathymetric ridges temporarily halt grounding line retreat. So we can conclude that bed topography is a major control on the rate of grounding line retreat.



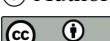


**Table 3.** Spearman correlation coefficients of determination between ice thickness and grounding line retreat rate

|      |             | Thwaites Glacier | Pine Island Glacier |
|------|-------------|:----------------:|:-------------------:|
| MISI | floatkill   | 0.06             | 0.74                |
| MICI | cmax 2 km/a | 0.11             | 0.62                |
|      | cmax 5 km/a | 0.15             | 0.61                |
|      | cmax 10 km/a| 0.20             | 0.50                |
|      | cmax 20 km/a| 0.07             | 0.60                |
|      | adaptive    | 0.04             | 0.01                |

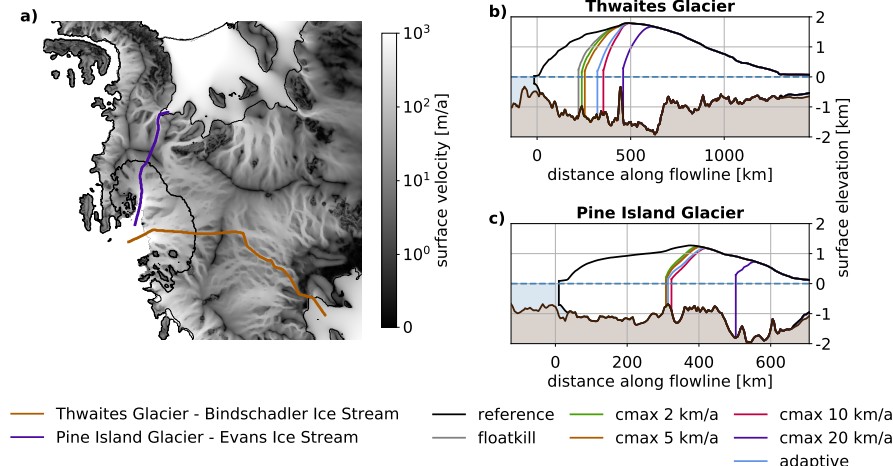

**Figure 7.** a) Map of flowlines from Pine Island glacier through Evans ice stream to Ronne ice shelf and from Thwaites glacier through Bindschadler ice stream to Ross ice shelf. b) and c) Bed topography and ice surface profiles after 60 a runtime for Thwaites glacier and Pine Island glacier, respectively. The distance along the flowline has its zero at the initial grounding line position. Note that for Pine Island glacier, the reference run also shows some grounding line retreat.

# 4 Processes and conditions that may slow or stop MICI

In this section we discuss our results in the light of mechanisms and conditions that may be important in limiting the speed of MICI evolution, including the influence of melange properties, climatic variations, and the ice/bed geometry.





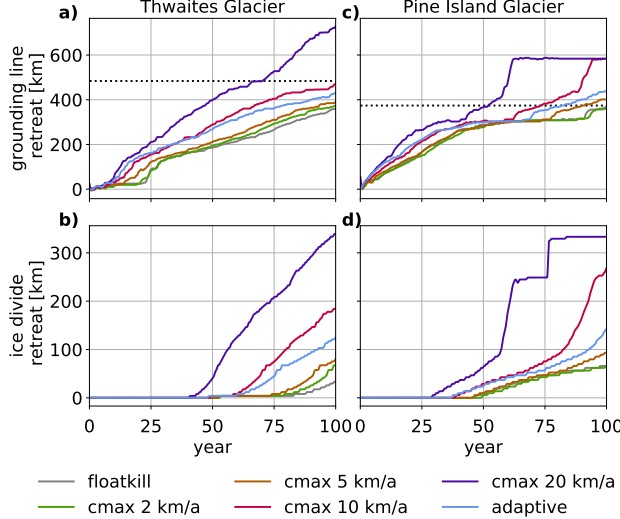

**Figure 8.** Grounding line retreat (a and c) and ice divide retreat (b and d) along the flowlines in Thwaites (a and b) and Pine Island glacier (c and d) as a function of simulated time. The dotted line shows the initial ice divide position.

## 4.1 The influence of mélange buttressing on calving rates and the speed of MICI

### 4.1.1 Realistic values for the mélange exit velocity

$C_{max}$ depends linearly on the embayment exit velocity $u_{ex}$ (see eq. 1). Therefore, constraining its range is important for estimating $C_{max}$: Maximum mélange flow speeds observed in front of Greenland glaciers are $30 - 50\,\text{m/d} \approx 10 - 18\,\text{km/a}$
(Amundson and Burton, 2018). The velocities of icebergs drifting in the Weddel Sea in Antarctica range from $9 - 15\,\text{km/d} \approx 3000 - 5500\,\text{km/a}$ (Schodlok et al., 2006). Since the exit velocity of mélange, $u_{ex}$, is expected to fall between mélange flow speed and iceberg drift velocity, we use an estimate of $u_{ex} = 100\,\text{km/a}$. However, smaller or larger values would also be consistent with observations.

For a given $u_{ex}$, the maximum possible $C_{max}$ is reached in the case of a wide and short embayment, $L_{em}/W_{em} \to 0$, which
provides little mélange buttressing due to its geometry. Assuming a rectangular embayment geometry with $W_{ex} = W_{cf}$, it is bounded by $C_{max} \le b_0^{-1}\gamma u_{ex}$. For $\gamma = 0.2$ and $u_{ex} = 100\,\text{km/a}$, this gives $C_{max} \le 17.1\,\text{km/a}$. Using a value of $u_{ex} = 10\,\text{km/a}$, which corresponds to the lower end of the observed mélange flow speeds, yields $C_{max} \le 1.7\,\text{km/a}$. This motivates the choice of values for $C_{max}$ in our experiments, ranging from $2 - 20\,\text{km/a}$.

### 4.1.2 Mélange build-up can stop MICI under winter conditions

Next, we investigate whether mélange buttressing can stop MICI after its onset. Under the assumption that no mélange leaves the embayment, calving is completely suppressed in the steady state ($u_{ex} = 0 \Rightarrow C_{max} = 0$, according to eq. 1). However,

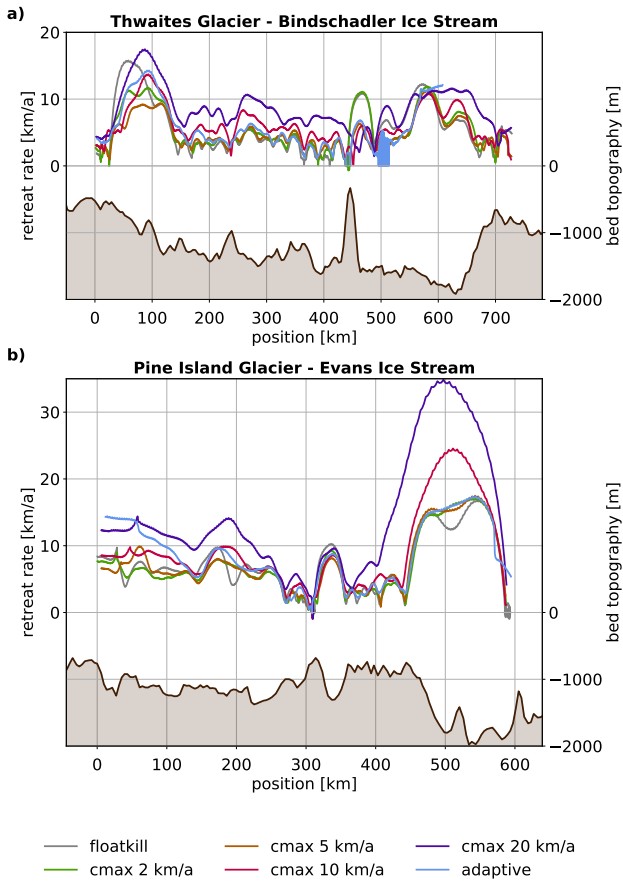

**Figure 9.** Grounding line retreat rates along the flowlines in Thwaites (a) and Pine Island glacier (b) as a function of grounding line position, together with bed topography.

when MICI begins, there may initially be little or no mélange in the embayment, therefore allowing calving until the mélange has reached a steady state.

We solve the non-steady state equations of the mélange-buttressing model as described in Schlemm and Levermann (2021), starting from a very thin mélange (10 m). Assuming that no mélange exits the embayment, mélange buildup can prevent calving almost completely within 10 a (see fig. 10a, grey lines).

In the mélange parametrization solved here, calving is suppressed completely if the mélange thickness at the glacier terminus, $d_{cf}$, reaches a given fraction of the ice thickness $H$: $d_{cf} \geq \gamma H$. However, studies explicitly analysing the influence of the mélange backpressure on the stress balance of the glacier terminus focus on the the force per unit width exerted by the mélange at the calving front (Amundson et al., 2010; Todd and Christoffersen, 2014; Crawford et al., 2021). Therefore, the force per unit width is calculated as a diagnostic variable. A mélange backpressure of $6.66 \cdot 10^6$ N/m is sufficient to prevent cliff calving





of an ice cliff with $H = 1000\,\mathrm{m}$ (Crawford et al., 2021). In our solution of the non-steady state equation, a similar force per unit width was found when calving is suppressed (see fig. 10c, grey lines after $> 5\,\mathrm{a}$).

If ice mélange freezes completely, no mélange exits the embayment. Mélange freezing and thereby stopping calving has been observed in Greenland glaciers in the winter season (Medrzycka et al., 2016). In the summer season, the sea ice in the mélange breaks up and the mélange becomes mobile. This seasonality can be modelled with a time-dependent mélange exit velocity of the form

$$u_{ex}(t) = u_0 \cdot \Big(1 + \arctan\big(k \cdot \sin(t \cdot 2\pi)\big)\Big)\big/\arctan(k) \quad \text{with } k = 20\,, \tag{3}$$

with a winter minimum of $u_{winter} = 0$, a summer maximum of $u_{summer} = 2u_0$ and an average of $u_0$. In the non-steady state
mélange model, this leads to seasonal variations in the strength of mélange buttressing (see fig. 10, orange and purple lines): After an initial equalibration period, mélange volume and backstress decrease in the summer and the calving rate increases, while in the winter mélange volume and backstress increase and the calving rate decreases. The minimum and maximum mélange properties fluctuate around the equilibrium value calculated by using the averaged exit velocity $u_0$. Contrary to observations, in this simplified mélange parameterization, winter freezing of mélange is not sufficient to stop calving. The reason is
that the equilibration of the mélange is too slow and takes several years rather than months or weeks.

   In conclusion, assuming that no mélange is lost by drifting off at the mélange exit, a very thick and strong mélange is built up within a period of several years, which completely prevents further calving and would thus stop the progression of MICI. However, this is only likely to happen in the winter season and would therefore halt MICI only temporarily.

### 4.1.3   Limitations of the idealized mélange buttressing parametrization

Due to its reliance on an idealized geometry, the mélange parametrization has several limitations when applied to realistic embayment geometries (see figs. 11a and b):

- The conversion of the realistic geometry into the idealized geometry is not unique: It is difficult to specify exactly where
each parameter of the idealized geometry should be measured.

- The mélange parameterization assumes a constant calving rate along the entire length of the calving front. This may be valid when considering a single glacier, but is no longer the case when several glaciers calve into the same embayment.

- On the west side of the Amundsen embayment, ice resting on bedrock above sea level forms pinning points that provide additional support to the ice mélange. This effect is neglected in the parameterization.

- The mélange length cannot be inferred from the model and must therefore be provided as an external parameter.

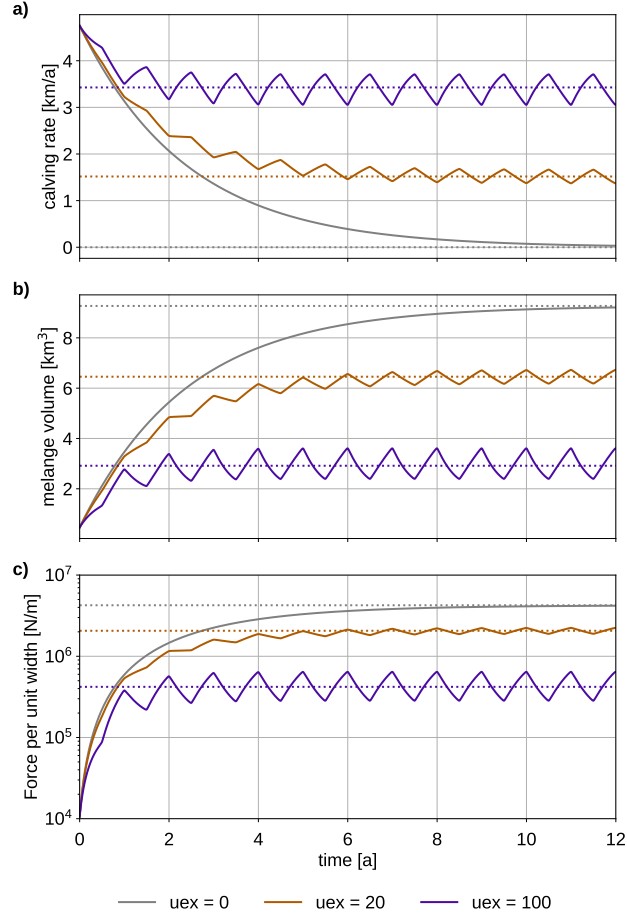

**Figure 10.** Evolution of the buttressed calving rate (a), the mélange volume (b) and the force per unit width at the calving front (c) in the case of no mélange exiting the embayment ($u_{ex} = 0$, grey lines) and for a seasonal variation in mélange exit velocity (orange and purple lines). The dotted lines show the corresponding equilibrium solution. The mélange geometry is rectangular with $W = 30$km, $L = 60$km, the initial mélange thickness at the calving front is $d_0 = 10$m and the unbuttressed calving rate is $C_0 = 5$km/a. For an equalibrated ice mélange, if no mélange exits the embayment ($u_{ex} = 0$), calving is completely suppressed ($C_{max} = 0$). However, in the time-dependent case and starting with a thin initial mélange, calving is possible for some years. Seasonal variations in the exit velocity lead to a seasonal variations of the mélange buttressing strength.

To get a better understanding of how mélange buttressing impacts calving rates in a realistic setup, it would be beneficial to use a spatially resolved mélange model. It should be able to handle realistic embayment geometries including pinning points as well as spatially resolved calving rates and have a criterion for where mélange stops being mélange, which would enable it to model mélange extent (see e.g. Pollard et al. (2018)). However, such a model introduces additional mélange parameters, 5 which are difficult to constrain.



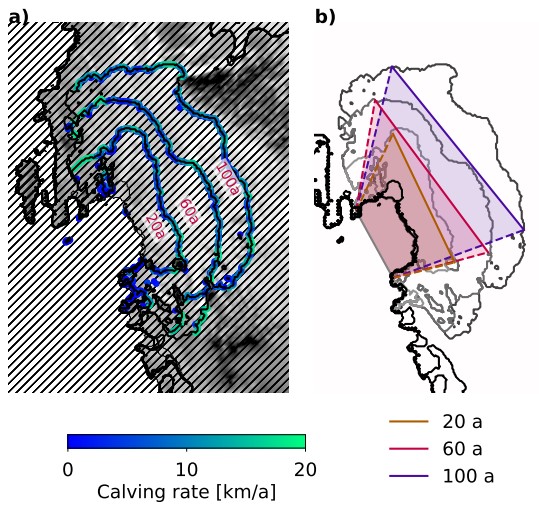

**Figure 11.** a) Different grounding line configurations of the adaptive cliff calving experiment with calving rates. b) Idealized embayment geometry derived from the grounding lines.

## 4.2 The role of ice shelves for MICI

The processes by which ice shelves fracture rapidly and disintegrate are not well enough understood to be implemented in an ice sheet model.

One way to remove ice shelves is by highly elevated basal melting. In PISM, this approach leaves small ice shelf remnants that

5    are only a few grid cells in size. The resulting buttressing loss induces MISI. However, since we assume that cliff calving only occurs at exposed, grounded ice cliffs, the ice shelf remnants prevent the onset of MICI. This is in contrast to the implementation in Pollard et al. (2015): They assumed that a small ice shelf remnant with vanishing buttressing strength does not prevent cliff calving, basing their reasoning on the Schoof flux across the grounding line (Schoof, 2007) and depth-averaged stresses in vicinity of the ice cliff (Bassis and Walker, 2011). However, the Schoof flux may not be applicable beyond a flowline setup

10    (Reese et al., 2018b). Additionally, a small ice shelf may impact the stress balance at the ice cliff in a 3d-setup. Therefore, we assume that cliff calving only occurs at exposed grounded ice cliffs.

In our model setup, we remove all floating ice in the Amundsen Basin and inner WAIS. This 'floatkill' parameterization mechanism eliminates all existing ice shelves at once in the first time step and prevents re-growth of ice shelves during the retreat. The removal of ice shelves initiates both MISI and MICI.

Two questions of vital importance for the onset and progress of MICI need further research:

1. Under which conditions do ice shelves collapse completely? Since ice shelf collapse is the prerequisite for the onset of MICI, the answer to this question determines when and if at all MICI will play a role for the future of the Antarctic ice sheet. There has been a lot of observational and theoretical work on hydrofracturing (MacAyeal et al., 2003; Robel and





Banwell, 2019) as well as rifting and crevassing (Borstad et al., 2012; Jeong et al., 2016; Lhermitte et al., 2020), but so far it is impossible to predict under which environmental and internal conditions a specific ice shelf will collapse.

2. Can ice shelves regrow after MICI has set in? If ice shelves can regrow after cliff calving has begun, this could stop MICI after its onset by buttressing the ice cliffs and preventing further cliff calving. However, if ice shelves cannot regrow, then MICI will continue mostly unhindered, because mélange buttressing can only slow the progress of MICI, but not stop it. Viscous deformation could prevent the formation of unstable ice cliffs (Clerc et al., 2019; Bassis et al., 2021) and allow ice shelves to regrow, whereas a mixed-mode behaviour of viscous deformation and fracture (Crawford et al., 2021) would make ice shelf regrowth unlikely.

## 4.3 Influence of regional climatic conditions on the progress of MICI

So far there are few observations of cliff calving glaciers. The retreat of Jakobshavn glacier in Greenland since 1998 (Joughin et al., 2008) was regarded as an indication that Jakobshavn glacier may be at the beginning of cliff calving regime (Bassis and Walker, 2011; DeConto and Pollard, 2016; Schlemm and Levermann, 2019). However, since 2016, Jacobshavn glacier has re-advanced as a result of regional ocean cooling (Khazendar et al., 2019). The cooling of the Fjord water has led to a decrease in frontal melt (Khazendar et al., 2019) as well as increased mélange buttressing at the glacier terminus (Joughin et al., 2020), thereby stopping its retreat. This suggests that changes in regional climatic conditions may slow or prevent grounding line retreat caused by cliff calving.

## 4.4 Slowdown of MICI at bathymetric ridges

During the last deglaciation, MICI was probably active for approximately 1000 a in the Amundsen region of the WAIS and then stopped, when the grounding line re-stabilized on a prominent bathymetric ridge (Wise et al., 2017). This is an indication that MICI can be stopped after its onset by features of the bed topography. However, our simulations show only temporary halts in grounding line retreat at bathymetric ridges in the interior of the WAIS (see fig. 9).

## 5 Conclusions

We performed PISM simulations of the WAIS to investigate the potential speeds of the two marine instabilities, MISI and MICI. We choose the Amundsen region as the starting point of the instabilities because observations show that MISI is possibly already in progress there. Due to ocean warming and increased crevassing, glaciers in the Amundsen region may lose their ice shelves in the future, which would set MICI in motion. We applied a 'floatkill' parameterization to remove the ice shelves in the Amundsen region, a cliff-calving parameterization depending on ice thickness, and a mélange-buttressing parameterization which limits calving rates.
We found that MISI, whether forced by the 'floatkill' parameterization or by high subshelf melt rates, has the potential to contribute $0.6\,\mathrm{m}$ of sea level rise within $100\,\mathrm{a}$. The sea level potential of MICI depends on the upper limit of calving: if the cliff calving rate is limited below $2\,\mathrm{km/a}$ or $5\,\mathrm{km/a}$, MICI has a smaller contribution to sea level rise than MISI. If the upper limit is $10\,\mathrm{km/a}$ or $20\,\mathrm{km/a}$, MICI doubles or even triples the sea level contribution of MISI.

We also showed that grounding line retreat is regulated by bed topography for both MISI and MICI. Although there is no clear statistical correlation between the retreat rate and the bed depth, we observe an accelerated retreat of the grounding line on retrograde beds and a slowdown on prograde beds.

Finally, we investigated how the upper limit for calving from mélange buttressing depends on the embayment geometry and the mélange exit velocity. Seasonal effects cause mélange build-up, which slows the progress of MICI temporarily under winter

condition. We also showed that as MICI progresses and the grounding line retreats, the calving front becomes longer while the width of the embayment exit remains the same. This leads to an increase in mélange buttressing, a decrease in the upper bound on calving rates, and consequently a slowdown in the progress of MICI. It is unlikely that mélange alone can completely stop MICI, but it could provide enough buttressing to enable ice shelf regrowth, which would then stop further MICI progress.

Future research is needed to gain a better understanding of the conditions under which MICI kicks off and to further constrain its potential sea level contribution.

The applied mélange parameterization assumes an idealized geometry and is therefore of limited applicability when extended to realistic embayment geometries. A spatially resolved mélange model might be a better choice. However, such a model would require more parameters describing mélange properties, which are difficult to constrain.

Two important unresolved questions about ice shelf collapse are beyond the scope of this study: first, under which conditions do ice shelves collapse? This determines the onset of MICI and is therefore crucially important in constraining at what degree of warming MICI becomes a concern. Second, can ice shelves regrow after MICI has started? This seems to be the only way to stop MICI. These two important questions control if and when MICI sets in and if it can be not only slowed down, but stopped completely after its onset.

*Code availability.* The code of PISM is openly available at https://pism-docs.org

*Code and data availability.* The code data will be made available upon publication.

*Competing interests.* We have no competing interests.



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
