# Peer review of "Stabilizing effect of mélange buttressing on the Marine Ice Cliff Instability of the West Antarctic Ice Sheet"

_The Cryosphere, 2021_

## Author Comment (AC1)

*General comments*

*This paper investigates the possible consequences of two main instabilities mechanisms (MISI and MICI) for the Amundsen region of the Antarctic Ice Sheet (AIS) using the Parallel Ice Sheet Model (PISM). The authors use a simple cliff-calving parameterization and a mélange buttressing model that were proposed and tested on idealized cases in their previous papers and apply them to real glacier cases in this paper. MICI is an important mechanism that could lead to large uncertainties in modeling the physics of the ice sheets. Recent theoretical and modeling studies (Ma et al., 2017, Bassis et al., 2017, Mercenier et al., 2018) have investigated and analyzed mechanisms for MICI. Applying a more physically based cliff-calving law on real glaciers setting is the timely step to makes an important contribution to the MICI hypothesis. However, I have several concerns described below that need to be addressed before publication. The impact and usefulness of the paper could be improved further if it were expanded to clarify some fundamental issues associated with constraining the idealized mélange buttressing parameterizations. Without observationally constrained models, the results would be over/under- estimating sea level contribution as other papers the authors mention in the manuscript.*

*1. Constraining the idealized mélange buttressing parameterization*

*The authors of course acknowledge and raise this issue in the manuscript. However, the application of the mélange buttressing parameterization is still limited because the model parameters are not observationally constrained. For example, the authors choose u_ex=100 km/a that falls between mélange flow speed (10-18 km/a) and iceberg drift velocity (3000-5500 km/a) but it's a quite large range (10~1000 km/a) and I think the authors arbitrarily choose this value (If I understand correctly). It seems that the results heavily depend on this u_ex value because it determines the upper bound of the calving rate (C_max) so constraining u_ex based on observations (e.g., observational calving rate) is key to applying the parameterization to real glacier cases. Please elaborate on how this u_ex is chosen and I suggest some sensitivity tests if needed.*

*Also, please elaborate on how other parameters for the cliff-calving law and mélange model are chosen or constrained with observations.*

**Reply:** The main parameter in the cliff calving parametrisation, C0, is poorly constrained because it depends on the timescale of shear failure and there are no experimental or observational studies for ice. However, in the mélange-buttressed case Cmax plays a much larger role in determining the calving rate, so the uncertainety in C0 does not matter very much (see Schlemm&Levermann 2021). In the melange buttressing parametrisation, mu0 is the coefficient of internal friction of the mélange and ranges from about 0.1 to larger than 1. Following Amundson and Burton (2018), we chose mu0=0.3. b0 and b1 follow from linearising an exponential relation from Amundson and Burton (2018) (see Schlemm&Levermann 2021). Details have been added to the manuscript.

In order to deal with the uncertainety in the melange exit velocity uex, we performed additionally a range of adaptive experiments with different uex values: 10 km/a (CCA10), 50 km/a (CCA50), 100 km/a as included in the original manuscript (CCA100), 200 km/a (CCA200), and 1000 km/a

(CCA1000). The resulting Cmax for the initial ice geometry lie between 1.5 km/a for CCA10 and 150 km/a for CCA1000. Consequently CCA10 shows very little cliff calving and only little more ice retreat than FLK or BMT. The Cmax of CCA1000 is initially so much larger than the actual cliff calving rate that there is effectively no melange buttressing. As the calving front retreats, Cmax decreases as in the CCA experiment included in the original manuscript. In contrast to the CC# experiments, where the upper bound Cmax was chosen adhoc, these experiments correlate an in principle observable parameter, the mélange exit velocity, to sea level rise. This does not actually constrain the uex value but gives a better understanding of its importance for the speed of MICI.

Amundson, J. M. and Burton, J. C.: Quasi-Static Granular Flow of Ice Mélange, J. Geophys. Res. Earth, 123, 2243–2257, https://doi.org/10.1029/2018JF004685, 2018.

Schlemm, T. and Levermann, A.: A simple parametrization of mélange buttressing for calving glaciers, The Cryosphere, 15, 531–545, https://doi.org/10.5194/tc-15-531-2021, 2021.

*2. Model description*

*I think the methods section is a little scant on details on model description. See below for more details.*

**Reply:** Details and references were added.

*P2L10: "This is referred to as Marine Ice Cliff Instability (MICI)": add references.*

**Reply:** Done.

*P4L6: "… at a horizontal resolution at 4 km". I think it would be beneficial if the authors could state the spatial and temporal resolution in a few sentences to make sure that results are numerically converged (i.e., independent of resolution) or the impact of resolution on the results.*

**Reply:** Done.

*P4L12: "The till friction angle is parameterized with bed elevation"*

*How is it parameterized? Please include details and references.*

**Reply:** Reference was added.

*P6L4: "The ice sheet was spun up into thermal equilibrium with fixed bed and ice geometry"*

*Please include details on how the model is initialized, for example, how long was the model spun up? What data (SMB, temperature) were used for this procedure? Also, include references if there are any.*

**Reply:** Length of the spinup was added. Boundary conditions (SMB, etc.) were previously described in section 2.1 and were moved into a separate section for more clarity.

*P6L9: "REF: a reference simulation with current day atmosphere and ocean conditions held constant"*

*What are the current day conditions for the atmosphere and ocean? Include details and references.*

**Reply:** Reference to the boundary conditions section was added.

*P7L1: "BMT: the 'basal melt experiment...200 m/a"*

*Why 200 m/a? any reference? Consider moving sentences from P8L18-L20 to this section.*

*What about the ice front? I think there is no calving in this experiment. Although it is explained later in the Results, please add how the ice front is dealt with (fixed or move) for this experiment.*

**Reply:** Description from P8L18-L20 was moved here. The ice front is free to evolve.

*P7L3: "FLK: the 'floatkill' ... removed"*

*Again, please add how the ice front is dealt with (fixed or move) for this experiment.*

**Reply:** Done.

*P8L2: "the three cliff calving experiments with small C_max".*

*Specify "small C_max". 2, 5, 10 km/a for C_max?*

**Reply:** Done.

*P8L1-3: "The 'floatkill' -parameterization... with C_max=20 km/a"*

*Where are the results of the extended simulations?*

**Reply:** The extended simulations are used for the flowline analysis.

*P8L31: "...reached the boundary of the inner WAIS region where cliff calving and the 'floatkill' parameterization are applied" -> floatkill parameterization are "not" applied.*

**Reply:** This was corrected.

*P10L2: "see fig3c" -> Figure 6?*

**Reply:** Reference is now to a table, in which Cmax values are given.

*P10L14: "This results in a slightly lower overall calving discharge"*

*What is that compared to? C_max=2 m/a without floatkill parameterization?*

*Does this issue with partially filled cells only show up when C_max=2 m/a? or also with C_max =5 m/a? Does this issue depend on the resolution of the domain? Please elaborate.*

**Reply:** Yes, this issue depends on the resolution of the domain. Previous unpublished sensitivity tests in a channel setup showed that for a resolution of x km, this problem occurs for calving rates smaller than x km/a. This has been added to the manuscript.

*P11L7: "...is shown in fig 3c." -> Fig. 6*

**Reply:** Reference is now to a table, in which Cmax values are given.

*P13L7: "...with the FLK experiment being the slowest, arriving there after 150a"*

*I don't see the results with extended time in the manuscript. Consider putting the results in the manuscript or appendix, or put "not shown here" in the text.*

**Reply:** "not shown here" was added in the manuscript.

*P16L7: "we use an estimate of u_ex=100 km/a. However, smaller or larger values would also be consistent with observations"*

*Why do you choose this value? Is the model calibrated against observation with this value? Please clarify what's consistent with observations. Have you done the sensitivity tests with u_ex values?*

**Reply:** New adaptive experiments were performed with a large range of values for uex and added to the manuscript.

*P16L15: "4.1.2 Melange build-up can stop MICI under winter conditions"*

*The title of this section could be misleading since the results show that mélange can stop MICI only if the winter condition (u_ex=0) lasts for several years, which is unlikely in real climate conditions.*

**Reply:** The title was corrected to "Winter freezing of mélange is not sufficient to stop MICI".

*P18L6: "This seasonality can be modelled with a time-dependent ..."*

*Does this experiment include melting/freezing of mélange? How are the results affected with melting/freezing of mélange?*

**Reply:** This model is described in more detail in Schlemm and Levermann (2021). It can describe melting of mélange, but no freezing. Therefore freezing is modelled by assuming a vanishing exit velocity. This was added to the manuscript.

*P18L26: "The mélange parameterization assumes a constant calving rate..."*

*Do you mean "the upper bound on calving rates (C_max)?*

**Reply:** The mélange parameterization assumes a constant calving rate along the entire length of the calving front. This has been clarified in the manuscript.

*P20L2: "The processes by which ice shelves fracture… in an ice sheet model"*

*Add references.*

**Reply:** Done.

*P22L14: "…but it could provide enough buttressing to enable ice shelf regrowth, which would then stop further MICI progress."*

*Is the ice shelf regrowth shown in model results or just from observations of Jakobshavn glacier?*

**Reply:** This is from observations of Jakobshavn. PISM allows shelves to regrow as soon as floatkill is turned off.

*Figure 4a: How is "calving discharge" calculated? Is this "(calving rate) x (area)" or ice discharge, that is "(velocity) X (area)"? If "calving discharge" is (calving rate) x (area), how is that calculated from the floatkill experiment which does not have calving? If calving discharge is ice discharge, the term "calving discharge" could be confusing.*

**Reply:** PISM uses a subgrid scheme at the ice margin where cells are partially filled with ice. In each time step, calving removes some of the ice in such a cell, whereas floatkill removes completely filled cell if they are floating. This removed ice volume is summed up in the calving discharge variable. This has been added to the manuscript.

*Figure 4b: The calving amplification value for C_max 20 km/a experiment seems larger than 6 but it's cut off. Please consider including the full extent if possible. Also, I am wondering why it suddenly increases toward the end of the simulation. From Figure 4a, the overall calving discharge for C_max 20 km/a looks highest near t=60a and decreases towards the end of the simulation.*

**Reply:** Plot ranges were adapted, so that all curves are visible. Since the new CCA1000 experiment has much higher values than the other experiments, an inset was used to show its whole range. The calving amplification of the CC20 and the CCA1000 experiments increases toward the end of the simulation time because parts of the grounding line have reached the margin of the inner WAIS region, beyond which cliff calving and the 'floatkill' mechanism are not applied. This has been added to the manuscript.

*Figure 5: Why do authors prohibit calving for the shaded area? Is it because of the bed topography>0 for that area? Include 0 m contour of bed topography since it's hard to see in the grey scale.*

**Reply:** The implementation of cliff calving in PISM allows no calving if the bed topography>0. The 0 m contour was added to the plot. However, if calving is allowed on all ice margins, ice

retreat starts all over Antarctica, for example on the Antarctic Peninsula. So calving is prohibited at all ice margins except in the Amundsen region. Because it is difficult to exclude only the margins, this prohibited region goes a little bit inland as a safety margin.

---

## Author Comment (AC2)

**Referee #3: Lizz Ultee**

*General comments*

*Schlemm et al investigate the progression and sea-level contribution potential of two proposed ice sheet instabilities in Antarctica. They use the Parallel Ice Sheet Model to simulate the retreat of the Amundsen Sea sector of the West Antarctic Ice Sheet at centennial scale, first in a "reference" case and then in seven experimental cases. The authors then compare sea-level contributions from the Marine Ice Cliff Instability (MICI) with those from the Marine Ice Sheet Instability (MISI) alone.*

*Overall, the manuscript raises interesting questions and presents a great deal of work toward addressing them. However, I found the organization confusing. Many results appear together with uneven levels of detail in the discussion. One of the major conclusions seems to be about interpreting an upper bound on cliff calving rate; the framing of that analysis in particular confused me. I would encourage the authors to carefully consider what key points they wish to highlight, and to thoroughly rework the manuscript to focus on those points.*

*I have self-cited in my specific comments below; I would prefer to avoid that but feel the work is relevant. I am therefore signing my review in the interest of transparency.*

*All the best,*

*Lizz Ultee*

**Specific comments**

*1. The authors find that their results depend strongly on what value is imposed as an "upper bound" on calving rate. This result makes intuitive sense. The authors also describe the calving rate parametrization as "loosely constrained". Here I am confused. There are two possible interpretations:*

- *Do the authors mean that the SLR contribution depends on the value of $C_{max}$ they impose in their simulations? That is certainly true, but does not make sense as a main result, because the authors have the ability to compute an adaptive $C_{max}$ that depends on local geometry. Results from an adaptive $C_{max}$ experiment are presented in Figure 3 and 4.*

- *Do the authors mean that the parameters in Equation 1 are loosely constrained? I think this is the more interesting point, and it could be more prominent in the text. The authors briefly discuss the value of the exit velocity $u_{ex}$ but do not further explore uncertainties in the adaptive calving rate. If related sensitivity studies have already been done, for example in Schlemm & Levermann 2021, it would be helpful to describe their key findings.*

*Given that the authors have already derived the adaptive bound and implemented it in a PISM experiment, I do not understand their focus on the four constant-$C_{max}$ experiments. Perhaps the constant-$C_{max}$ experiments are intended to provide context for the range of possibilities of a poorly constrained adaptive $C_{max}$? If so, I need more framing from the authors to aid that interpretation.*

*The authors are right to point out the limitations in DeConto & Pollard's (2016) approach, which imposes a very fast cliff calving retreat rate. However, in my view, applying various constant cliff*

*calving rates as upper bounds is not much improvement on DeConto & Pollard (2016). For one thing, the theoretical upper bound on cliff calving rate depends on outlet glacier geometry, local climate, and the yield strength of ice (Bassis & Ultee 2019; implemented for all grounded GrIS outlets in Ultee & Bassis 2020). Secondly, the authors themselves have derived a parametrization for melange-buttressed cliff calving, which brings in more interesting physics than the constant rates.*

*One way to reorganize might be to focus more directly on the adaptive calving rate experiment in the main text, and separate the constant-rate experiments into a supplement or even into their own "brief communication" style manuscript. I don't insist on this—simply a suggestion for how a more focused manuscript might read.*

**Reply:** Thank you for the suggestion. In addition to the cliff calving experiments with different Cmax values (CC#), we performed a range of adaptive experiments with different uex values: 10 km/a (CCA10), 50 km/a (CCA50), 100 km/a as included in the original manuscript (CCA100), 200 km/a (CCA200), and 1000 km/a (CCA1000). The resulting Cmax for the initial ice geometry lie between 1.5 km/a for CCA10 and 150 km/a for CCA1000. Consequently CCA10 shows very little cliff calving and only little more ice retreat than FLK or BMT. The Cmax of CCA1000 is initially so much larger than the actual cliff calving rate that there is effectively no melange buttressing. As the calving front retreats, Cmax decreases significantly as in the CCA experiment included in the original manuscript. In contrast to the CC# experiments, where the upper bound Cmax was chosen adhoc, these experiments correlate an in principle observable parameter, the mélange exit velocity, to sea level rise. This does not actually constrain the uex value but gives a better understanding of its importance for the speed of MICI.

*2. I could use more explanation of the assumptions of the $C_{max}$ bound. The authors write on p5, L22: "In order to estimate Cmax for a given grounding line configuration, we assume that the entire embayment is filled with mélange." I do not understand how this is an upper bound on calving rate. An embayment entirely filled with mélange would be the configuration that results in the most suppression of calving, but couldn't there be \*more\* calving if the embayment were not filled with mélange?*

**Reply:** Yes, there would be more calving, if the embayment was not filled with mélange. However the mélange parametrisation cannot evolve the mélange margin, we need to assume its position. Evolving mélange thickness can be modelled though: if the whole embayment is filled with very thin, spread-out mélange, the calving rate is large and many icebergs are produced. As a result the mélange thickness grows quickly and reaches its equilibrium thickness within a few years (see Schlemm&Levermann 2021). This has been added to the manuscript.

*3. The experiments presented in Section 3.4 allow the ice divides and grounding lines of Thwaites and PIG to retreat all the way across WAIS to the Ross and Ronne-Filchner Ice Shelves. How realistic is this? In such cases, I'd expect some lateral motion of the ice divides and resulting adjustment in the grounding lines. It would be helpful to have more guidance from the authors in interpreting the real-world context of these results.*

**Reply:** These are the same 2d-experiments discussed in the rest of the paper, just analysed along the trajectory of the flowlines. The ice divides are free to move. It may be that, due to movement of the ice divide, the actual flowline, aka main direction of the ice flow, changes. This has not been taken into account. This has been clarified in the manuscript.

*4. The authors describe seasonal variations in the strength of mélange buttressing in Section 4.1.2. I have two concerns with this. First, the results are buried much later in the manuscript than other primary results of experiments. I was confused to see them there. Second, the title of the section reads "Mélange build-up can stop MICI under winter conditions", and p16, L16 indicates that result, but p18, L14 reads "…winter freezing of mélange is not sufficient to stop calving." I am left unsure whether frozen mélange inhibits MICI or not.*

**Reply:** This section was split up and moved forwards, it is now included in the method section as well as in the results section. The title was corrected to "Winter freezing of mélange is not sufficient to stop MICI".

*Technical corrections*

*P5, L8: "can lead to very large calving rates" - please clarify whether this refers to large calving rates in the model or in observations*

**Reply:** It refers to the model, this has been clarified.

*P10, L13-16: "Cliff calving with a small value of $C_{max}$…in this case." I do not understand this explanation. Consider rephrasing or elaborating.*

**Reply:** This has been clarified.

*P11, L3-4: "Because the embayment becomes wider…the upper bound on calving rate decreases with grounding line retreat into the Amundsen basin." This does not agree with the decline in buttressing that I interpret from Figure 2 and previous description. Please check this and/or clarify the explanation.*

**Reply:** The distance between the calving front and the embayment exit increases. This has been corrected.

*Figures 3 & 4: Please consider using thicker lines for the legend entries, or labelling $C_{max}$ directly on the plot. I find the colors hard to distinguish with the thin lines currently in the legend.*

**Reply:** Done.

*Section 4.3: The official name of Greenland's fastest outlet, formerly called Jakobshavn Isbræ, is Sermeq Kujalleq (Bjørk, Kruse & Michaelsen 2015). Please update the nomenclature you use to discuss it.*

**Reply:** Thanks for pointing this out, we corrected it.

---

## Author Comment (AC3)

**Response to comments by Jeremy Bassis**

*A theme that has been helpful for many of these past debates about physical processes, rests in asking what observations can be used to test the model or better constrain the parameters? I personally think that this link back to existing or needed observations is a useful question to expound upon in the manuscript. How do we know the model (or certain parameter combinations) are capturing some of the relevant physics. Are there specific predictions that the model can make that can be tested to (in)validate any of the model hypotheses?*

**Reply:** We agree that this is an important point. How might the predictions of the mélange-buttressed cliff-calving parametrisation be tested against observations? There is limited data on Antarctic glaciers and most of them still have shelves, so they are not currently in a cliff-calving regime. In Greenland, however, most glaciers terminate in narrow fjords filled with mélange and many have no floating tongue – they may therefore be subject to cliff calving. The cliff calving parametrization was confirmed as a lower limit on calving rates for Sermeq Kujalleq (Jakobshavn glacier) in Schlemm & Levermann (2019). This is because the cliff-calving parametrisation used here underestimates calving rates, if the glacier is just at the beginning of the cliff-calving regime. This is in contrast to Ultee & Bassis (2020), where a calving model based on a thin film viscoplastic theory was shown to provide an upper limit for calving rates of Greenland glaciers. Since the cliff-calving parametrisation underestimates calving rates of current glaciers, another calving parametrisation would need to be used in order to test the mélange buttressing parametrisations against observations. However, then it would be unclear how to differentiate between fitting the calving model and fitting the mélange model.

Ultee & Bassis (2020). SERMeQ model produces a realistic upper bound on calving retreat for 155 Greenland outlet glaciers. *Geophysical Research Letters, 47*(21): e2020GL090213. https://doi.org/10.1029/2020GL090213

Schlemm, T. & Levermann, A (2019). A simple stress-based cliff-calving law. *The Cryosphere*, 13, 2475-2488

*Another important issue is one of numerical convergence. My view has long been that modelers (I include myself in this) need to more systematically demonstrate numerical convergence of models, but we often forget or assume that because it was done for a previous study that it doesn't need to be repeated. I do think that a numerical convergence study, even if it is for a shorter period of time or limited portion of the domain would better allow the authors to demonstrate how robust the results are to numerical parameters.*

**Reply:** We agree. We know that the model is not converged at a resolution of 4km. However using the subgrid scheme at the grounding line, the reversibility of the grounding line is captured well as shown in Feldmann et. al. (2014).

Feldmann, J.; Albrecht, T.; Khroulev, C.; Pattyn, F. & Levermann, A. (2014). Resolution-dependent performance of grounding line motion in a shallow model compared with a full-Stokes model according to the MISMIP3d intercomparison. *Journal of Glaciology, Cambridge University Press,* 2014, 60, 353–360

---

## Editor Decision (ED1)

[revised manuscript text omitted]

To better constrain the MICI in Antarctica, it would be desirable to test the predictions of the melange-buttressed cliff-calving

20   parameterization against observations. However, there are few data on Antarctic glaciers, and most of them still have shelves, so they are not currently in a cliff-calving regime. In Greenland, on the other hand, most glaciers terminate in narrow mélange-filled fjords, and many have no floating tongue: they may therefore be subject to cliff calving. The cliff calving parameterization was supported as a lower bound on calving rates for Sermeq Kujalleq (Jakobshavn Glacier) in (Schlemm and Levermann, 2019). This is because it underestimates calving rates when the glacier is just at the beginning of the cliff calving regime.

25   This is in contrast to Ultee and Bassis (2020), where it was shown that a calving model based on a viscoplastic thin-sheet theory provides an upper bound on the calving rates of Greenland glaciers. Because the cliff calving parameterization used here underestimates the calving rates of present-day glaciers, a different calving parameterization would need to be used to test the melange-buttressing parameterization against observations. However, it would then be unclear how to distinguish between fitting the calving model and fitting the melange model.

[revised manuscript text omitted]

35  Ultee, L. and Bassis, J. N.: SERMeQ Model Produces a Realistic Upper Bound on Calving Retreat for 155 Greenland Outlet Glaciers, Geophysical Research Letters, 47, e2020GL090 213, https://doi.org/https://doi.org/10.1029/2020GL090213, https://agupubs.onlinelibrary.wiley.com/doi/abs/10.1029/2020GL090213, e2020GL090213 10.1029/2020GL090213, 2020.

van Wessem, J. M., van de Berg, W. J., Noël, B. P. Y., van Meijgaard, E., Amory, C., Birnbaum, G., Jakobs, C. L., Krüger, K., Lenaerts, J. T. M., Lhermitte, S., Ligtenberg, S. R. M., Medley, B., Reijmer, C. H., van Tricht, K., Trusel, L. D., van Ulft, L. H., Wouters, B., Wuite, J., and van den Broeke, M. R.: Modelling the climate and surface mass balance of polar ice sheets using RACMO2 – Part 2: Antarctica (1979–2016), The Cryosphere, 12, 1479–1498, https://doi.org/10.5194/tc-12-1479-2018, https://tc.copernicus.org/articles/12/1479/2018/, 2018.

WCRP Global Sea Level Budget Group: Global sea-level budget 1993–present, Earth System Science Data, 10, 1551–1590, https://doi.org/10.5194/essd-10-1551-2018, https://www.earth-syst-sci-data.net/10/1551/2018/, 2018.

Winkelmann, R., Martin, M. A., Haseloff, M., Albrecht, T., Bueler, E., Khroulev, C., and Levermann, A.: The Potsdam Parallel Ice Sheet Model (PISM-PIK) – Part 1: Model description, The Cryosphere, 5, 715–726, https://doi.org/10.5194/tc-5-715-2011, http://www.the-cryosphere.net/5/715/2011/, 2011.

Wise, M. G., Dowdeswell, J. A., Jakobsson, M., and Larter, R. D.: Evidence of marine ice-cliff instability in Pine Island Bay from iceberg-keel plough marks, Nature, 550, 506–510, https://doi.org/10.1038/nature24458, 2017.

Yu, H., Rignot, E., Morlighem, M., and Seroussi, H.: Iceberg calving of Thwaites Glacier, West Antarctica: full-Stokes modeling combined with linear elastic fracture mechanics, The Cryosphere, 11, 1283–1296, https://doi.org/10.5194/tc-11-1283-2017, https://tc.copernicus.org/articles/11/1283/2017/, 2017.

Zwally, H. J., Giovinetto, M. B., Beckley, M. A., and Sab, J. L.: Antarctic and Greenland Drainage Systems, http://icesat4.gsfc.nasa.gov/cryo_data/ant_grn_drainage_systems.php, 2012.

---

## Author Response (AR2)

Reply to the comments:

All requested changes were made. The plots with insets were updated with an opaque background of the inset. Coda and data availability as well as authors' contribution were updated.